# End to end learning and optimization on graphs

**Bryan Wilder**
Harvard University
bwilder@g.harvard.edu

**Eric Ewing**
University of Southern California
ericewin@usc.edu

**Bistra Dilkina**
University of Southern California
dilkina@usc.edu

**Milind Tambe**
Harvard University
milind_tambe@harvard.edu

## Abstract

Real-world applications often combine learning and optimization problems on graphs. For instance, our objective may be to cluster the graph in order to detect meaningful communities (or solve other common graph optimization problems such as facility location, maxcut, and so on). However, graphs or related attributes are often only partially observed, introducing learning problems such as link prediction which must be solved prior to optimization. Standard approaches treat learning and optimization entirely separately, while recent machine learning work aims to predict the optimal solution directly from the inputs. Here, we propose an alternative *decision-focused learning* approach that integrates a differentiable proxy for common graph optimization problems as a layer in learned systems. The main idea is to learn a representation that maps the original optimization problem onto a simpler proxy problem that can be efficiently differentiated through. Experimental results show that our CLUSTERNET system outperforms both pure end-to-end approaches (that directly predict the optimal solution) and standard approaches that entirely separate learning and optimization. Code for our system is available at https://github.com/bwilder0/clusternet.

## 1 Introduction

While deep learning has proven enormously successful at a range of tasks, an expanding area of interest concerns systems that can flexibly combine learning with optimization. Examples include recent attempts to solve combinatorial optimization problems using neural architectures [45, 28, 8, 30], as well as work which incorporates explicit optimization algorithms into larger differentiable systems [3, 18, 47]. The ability to combine learning and optimization promises improved performance for real-world problems which require decisions to be made on the basis of machine learning predictions by enabling end-to-end training which focuses the learned model on the decision problem at hand.

We focus on graph optimization problems, an expansive subclass of combinatorial optimization. While graph optimization is ubiquitous across domains, complete applications must also solve machine learning challenges. For instance, the input graph is usually incomplete; some edges may be unobserved or nodes may have attributes that are only partially known. Recent work has introduced sophisticated methods for tasks such as link prediction and semi-supervised classification [38, 29, 39, 25, 53], but these methods are developed in isolation of downstream optimization tasks. Most current solutions use a two-stage approach which first trains a model using a standard loss and then plugs the model's predictions into an optimization algorithm ([50, 10, 5, 9, 42]). However, predictions which minimize a standard loss function (e.g., cross-entropy) may be suboptimal for specific optimization tasks, especially in difficult settings where even the best model is imperfect.

A preferable approach is to incorporate the downstream optimization problem into the training of the machine learning model. A great deal of recent work takes a pure end-to-end approach where a neural network is trained to predict a solution to the optimization problem using supervised or reinforcement learning [45, 28, 8, 30]. However, this often requires a large amount of data and results in suboptimal performance because the network needs to discover algorithmic structure entirely from scratch. Between the extremes of an entirely two stage approach and pure end-to-end architectures, *decision-focused learning* [18, 47] embeds a solver for the optimization problem as a differentiable layer within a learned system. This allows the model to train using the downstream performance that it induces as the loss, while leveraging prior algorithmic knowledge for optimization. The downside is that this approach requires manual effort to develop a differentiable solver for each particular problem and often results in cumbersome systems that must, e.g, call a quadratic programming solver every forward pass.

We propose a new approach that gets the best of both worlds: incorporate a solver for a simpler optimization problem as a differentiable layer, and then learn a representation that maps the (harder) problem of interest onto an instance of the simpler problem. Compared to earlier approaches to decision-focused learning, this places more emphasis on the representation learning component of the system and simplifies the optimization component. However, compared to pure end-to-end approaches, we only need to learn the reduction to the simpler problem instead of the entire algorithm.

In this work, we instantiate the simpler problem as a differentiable version of $k$-means clustering. Clustering is motivated by the observation that graph neural networks embed nodes into a continuous space, allowing us to approximate optimization over the discrete graph with optimization in continuous embedding space. We then interpret the cluster assignments as a solution to the discrete problem. We instantiate this approach for two classes of optimization problems: those that require *partitioning* the graph (e.g., community detection or maxcut), and those that require *selecting a subset of $K$ nodes* (facility location, influence maximization, immunization, etc). We don't claim that clustering is the right algorithmic structure for all tasks, but it is sufficient for many problems as shown in this paper.

In short, we make three contributions. First, we introduce a general framework for integrating graph learning and optimization, with a simpler optimization problem in continuous space as a proxy for the more complex discrete problem. Second, we show how to differentiate through the clustering layer, allowing it to be used in deep learning systems. Third, we show experimental improvements over both two-stage baselines as well as alternate end-to-end approaches on a range of example domains.

## 2   Related work

We build on a recent work on decision-focused learning [18, 47, 15], which includes a solver for an optimization problem into training in order to improve performance on a downstream decision problem. A related line of work develops and analyzes effective surrogate loss functions for predict-then-optimize problems [19, 6]. Some work in structured prediction also integrates differentiable solvers for discrete problems (e.g., image segmentation [16] or time series alignment [34]). Our work differs in two ways. First, we tackle more difficult optimization problems. Previous work mostly focuses on convex problems [18] or discrete problems with near-lossless convex relations [47, 16]. We focus on highly combinatorial problems where the methods of choice are hand-designed discrete algorithms. Second, in response to this difficulty, we differ methodologically in that we do not attempt to include a solver for the exact optimization problem at hand (or a close relaxation of it). Instead, we include a more generic algorithmic skeleton that is automatically finetuned to the optimization problem at hand.

There is also recent interest in training neural networks to solve combinatorial optimization problems [45, 28, 8, 30]. While we focus mostly on combining graph learning with optimization, our model can also be trained just to solve an optimization problem given complete information about the input. The main methodological difference is that we include more structure via a differentiable $k$-means layer instead of using more generic tools (e.g., feed-forward or attention layers). Another difference is that prior work mostly trains via reinforcement learning. By contrast, we use a differentiable approximation to the objective which removes the need for a policy gradient estimator. This is a benefit of our architecture, in which the final decision is fully differentiable in terms of the model parameters instead of requiring non-differentiable selection steps (as in [28, 8, 30]). We give our

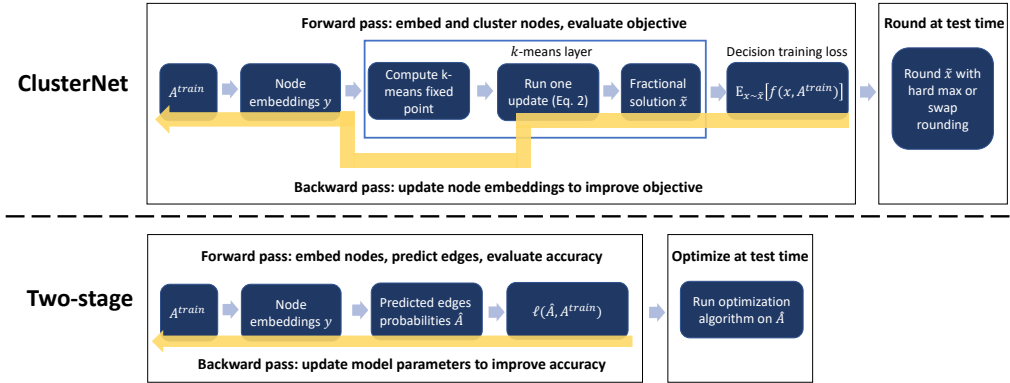

Figure 1: Top: CLUSTERNET, our proposed system. Bottom: a typical two-stage approach.

end-to-end baseline ("GCN-e2e") the same advantage by training it with the same differentiable decision loss as our own model instead of forcing it to use noisier policy gradient estimates.

Finally, some work uses deep architectures as a part of a clustering algorithm [43, 31, 24, 41, 35], or includes a clustering step as a component of a deep network [21, 22, 52]. While some techniques are similar, the overall task we address and framework we propose are entirely distinct. Our aim is not to cluster a Euclidean dataset (as in [43, 31, 24, 41]), or to solve perceptual grouping problems (as in [21, 22]). Rather, we propose an approach for graph optimization problems. Perhaps the closest of this work is Neural EM [22], which uses an unrolled EM algorithm to learn representations of visual objects. Rather than using EM to infer representations for objects, we use $k$-means in graph embedding space to solve an optimization problem. There is also some work which uses deep networks for graph clustering [49, 51]. However, none of this work includes an explicit clustering algorithm in the network, and none consider our goal of integrating graph learning and optimization.

## 3 Setting

We consider settings that combine learning and optimization. The input is a graph $G = (V, E)$, which is in some way partially observed. We will formalize our problem in terms of link prediction as an example, but our framework applies to other common graph learning problems (e.g., semi-supervised classification). In link prediction, the graph is not entirely known; instead, we observe only training edges $E^{train} \subset E$. Let $A$ denote the adjacency matrix of the graph and $A^{train}$ denote the adjacency matrix with only the training edges. The learning task is to predict $A$ from $A^{train}$. In domains we consider, the motivation for performing link prediction, is to solve a decision problem for which the objective depends on the full graph. Specifically, we have a decision variable $x$, objective function $f(x, A)$, and a feasible set $\mathcal{X}$. We aim to solve the optimization problem

$$\max_{x \in \mathcal{X}} f(x, A). \tag{1}$$

However, $A$ is unobserved. We can also consider an inductive setting in which we observe graphs $A_1, ..., A_m$ as training examples and then seek to predict edges for a partially observed graph from the same distribution. The most common approach to either setting is to train a model to reconstruct $A$ from $A^{train}$ using a standard loss function (e.g., cross-entropy), producing an estimate $\hat{A}$. The *two-stage* approach plugs $\hat{A}$ into an optimization algorithm for Problem 1, maximizing $f(x, \hat{A})$.

We propose end-to-end models which map from $A^{train}$ directly to a feasible decision $x$. The model will be trained to maximize $f(x, A^{train})$, i.e., the quality of its decision evaluated on the training data (instead of a loss $\ell(\hat{A}, A^{train})$ that measures purely predictive accuracy). One approach is to "learn away" the problem by training a standard model (e.g., a GCN) to map directly from $A^{train}$ to $x$. However, this forces the model to entirely rediscover algorithmic concepts, while two-stage methods are able to exploit highly sophisticated optimization methods. We propose an alternative that embeds algorithmic structure into the learned model, getting the best of both worlds.

# 4 Approach: CLUSTERNET

Our proposed CLUSTERNET system (Figure 1) merges two differentiable components into a system that is trained end-to-end. First, a *graph embedding* layer which uses $A^{train}$ and any node features to embed the nodes of the graph into $\mathbb{R}^p$. In our experiments, we use GCNs [29]. Second, a layer that performs *differentiable optimization*. This layer takes the continuous-space embeddings as input and uses them to produce a solution $x$ to the graph optimization problem. Specifically, we propose to use a layer that implements a differentiable version of $K$-means clustering. This layer produces a soft assignment of the nodes to clusters, along with the cluster centers in embedding space.

The intuition is that cluster assignments can be interpreted as the solution to many common graph optimization problems. For instance, in community detection we can interpret the cluster assignments as assigning the nodes to communities. Or, in maxcut, we can use two clusters to assign nodes to either side of the cut. Another example is maximum coverage and related problems, where we attempt to select a set of $K$ nodes which cover (are neighbors to) as many other nodes as possible. This problem can be approximated by clustering the nodes into $K$ components and choosing nodes whose embedding is close to the center of each cluster. We do not claim that any of these problems is exactly reducible to $K$-means. Rather, the idea is that including $K$-means as a layer in the network provides a useful inductive bias. This algorithmic structure can be fine-tuned to specific problems by training the first component, which produces the embeddings, so that the learned representations induce clusterings with high objective value for the underlying downstream optimization task. We now explain the optimization layer of our system in greater detail. We start by detailing the forward and the backward pass for the clustering procedure, and then explain how the cluster assignments can be interpreted as solutions to the graph optimization problem.

## 4.1 Forward pass

Let $x_j$ denote the embedding of node $j$ and $\mu_k$ denote the center of cluster $k$. $r_{jk}$ denotes the degree to which node $j$ is assigned to cluster $k$. In traditional $K$-means, this is a binary quantity, but we will relax it to a fractional value such that $\sum_k r_{jk} = 1$ for all $j$. Specifically, we take $r_{jk} = \frac{\exp(-\beta||x_j - \mu_k||)}{\sum_\ell \exp(-\beta||x_j - \mu_\ell||)}$, which is a soft-min assignment of each point to the cluster centers based on distance. While our architecture can be used with any norm $||\cdot||$, we use the negative cosine similarity due to its strong empirical performance. $\beta$ is an inverse-temperature hyperparameter; taking $\beta \to \infty$ recovers the standard $k$-means assignment. We can optimize the cluster centers via an iterative process analogous to the typical $k$-means updates by alternately setting

$$\mu_k = \frac{\sum_j r_{jk} x_j}{\sum_j r_{jk}} \; \forall k = 1...K \quad r_{jk} = \frac{\exp(-\beta||x_j - \mu_k||)}{\sum_\ell \exp(-\beta||x_j - \mu_\ell||)} \; \forall k = 1...K, j = 1...n. \quad (2)$$

These iterates converge to a fixed point where $\mu$ remains the same between successive updates [33]. The output of the forward pass is the final pair $(\mu, r)$.

## 4.2 Backward pass

We will use the implicit function theorem to analytically differentiate through the fixed point that the forward pass $k$-means iterates converge to, obtaining expressions for $\frac{\partial \mu}{\partial x}$ and $\frac{\partial r}{\partial x}$. Previous work [18, 47] has used the implicit function theorem to differentiate through the KKT conditions of optimization problems; here we take a more direct approach that characterizes the update process itself. Doing so allows us to backpropagate gradients from the decision loss to the component that produced the embeddings $x$. Define a function $f : \mathbb{R}^{Kp} \to \mathbb{R}$ as

$$f_{i,\ell}(\mu, x) = \mu_i^\ell - \frac{\sum_j r_{jk} x_j^\ell}{\sum_j r_{jk}} \quad (3)$$

Now, $(\mu, x)$ are a fixed point of the iterates if $f(\mu, x) = \mathbf{0}$. Applying the implicit function theorem yields that $\frac{\partial \mu}{\partial x} = - \left[ \frac{\partial f(\mu, x)}{\partial \mu} \right]^{-1} \frac{\partial f(\mu, x)}{\partial x}$, from which $\frac{\partial r}{\partial x}$ can be easily obtained via the chain rule.

**Exact backward pass:** We now examine the process of calculating $\frac{\partial \mu}{\partial x}$. Both $\frac{\partial f(\mu, x)}{\partial x}$ and $\frac{\partial f(\mu, x)}{\partial \mu}$ can be easily calculated in closed form (see appendix). Computing the former requires time $O(nKp^2)$.

Computing the latter requires $O(npK^2)$ time, after which it must be inverted (or else iterative methods must be used to compute the product with its inverse). This requires time $O(K^3p^3)$ since it is a matrix of size $(Kp) \times (Kp)$. While the exact backward pass may be feasible for some problems, it quickly becomes burdensome for large instances. We now propose a fast approximation.

**Approximate backward pass:** We start from the observation that $\frac{\partial f}{\partial \mu}$ will often be dominated by its diagonal terms (the identity matrix). The off-diagonal entries capture the extent to which updates to one entry of $\mu$ indirectly impact other entries via changes to the cluster assignments $r$. However, when the cluster assignments are relatively firm, $r$ will not be highly sensitive to small changes to the cluster centers. We find to be typical empirically, especially since the optimal choice of the parameter $\beta$ (which controls the hardness of the cluster assignments) is typically fairly high. Under these conditions, we can approximate $\frac{\partial f}{\partial \mu}$ by its diagonal, $\frac{\partial f}{\partial \mu} \approx I$. This in turn gives $\frac{\partial \mu}{\partial x} \approx -\frac{\partial f}{\partial x}$.

We can formally justify this approximation when the clusters are relatively balanced and well-separated. More precisely, define $c(j) = \arg\max_i r_{ji}$ to be the closest cluster to point $j$. Proposition 1 (proved in the appendix) shows that the quality of the diagonal approximation improves exponentially quickly in the product of two terms: $\beta$, the hardness of the cluster assignments, and $\delta$, which measures how well separated the clusters are. $\alpha$ (defined below) measures the balance of the cluster sizes. We assume for convenience that the input is scaled so $||x_j||_1 \leq 1 \, \forall j$.

**Proposition 1.** *Suppose that for all points $j$, $||x_j - \mu_i|| - ||x_j - \mu_{c(j)}|| \geq \delta$ for all $i \neq c(j)$ and that for all clusters $i$, $\sum_{j=1}^{n} r_{ji} \geq \alpha n$. Moreover, suppose that $\beta\delta > \log \frac{2\beta K^2}{\alpha}$. Then, $\left\| \frac{\partial f}{\partial \mu} - I \right\|_1 \leq \exp(-\delta\beta) \left( \frac{K^2\beta}{\frac{1}{2}\alpha - K^2\beta \exp(-\delta\beta)} \right)$ where $|| \cdot ||_1$ is the operator 1-norm.*

We now show that the approximate gradient obtained by taking $\frac{\partial f}{\partial \mu} = I$ can be calculated by unrolling a single iteration of the forward-pass updates from Equation 2 at convergence. Examining Equation 3, we see that the first term ($\mu_i^\ell$) is constant with respect to $x$, since here $\mu$ is a fixed value. Hence,

$$-\frac{\partial f_k}{\partial x} = \frac{\partial}{\partial x} \frac{\sum_j r_{jk} x_j}{\sum_j r_{jk}}$$

which is just the update equation for $\mu_k$. Since the forward-pass updates are written entirely in terms of differentiable functions, we can automatically compute the approximate backward pass with respect to $x$ (i.e., compute products with our approximations to $\frac{\partial \mu}{\partial x}$ and $\frac{\partial r}{\partial x}$) by applying standard autodifferentiation tools to the final update of the forward pass. Compared to computing the exact analytical gradients, this avoids the need to explicitly reason about or invert $\frac{\partial f}{\partial \mu}$. The final iteration (the one which is differentiated through) requires time $O(npK)$, *linear* in the size of the data.

Compared to differentiating by unrolling the entire sequence of updates in the computational graph (as has been suggested for other problems [17, 4, 54]), our approach has two key advantages. First, it avoids storing the entire history of updates and backpropagating through all of them. The runtime for our approximation is independent of the number of updates needed to reach convergence. Second, *we can in fact use entirely non-differentiable operations to arrive at the fixed point*, e.g., heuristics for the $K$-means problem, stochastic methods which only examine subsets of the data, etc. This allows the forward pass to scale to larger datasets since we can use the best algorithmic tools available, not just those that can be explicitly encoded in the autodifferentiation tool's computational graph.

## 4.3 Obtaining solutions to the optimization problem

Having obtained the cluster assignments $r$, along with the centers $\mu$, in a differentiable manner, we need a way to **(1)** differentiably interpret the clustering as a soft solution to the optimization problem, **(2)** differentiate a relaxation of the objective value of the graph optimization problem in terms of that solution, and then **(3)** round to a discrete solution at test time. We give a generic means of accomplishing these three steps for two broad classes of problems: those that involve *partitioning the graph into $K$ disjoint components*, and those that that involve *selecting a subset of $K$ nodes*.

**Partitioning:** **(1)** We can naturally interpret the cluster assignments $r$ as a soft partitioning of the graph. **(2)** One generic continuous objective function (defined on soft partitions) follows from the random process of assigning each node $j$ to a partition with probabilities given by $r_j$, repeating this process independently across all nodes. This gives the expected training decision loss

Table 1: Performance on the community detection task

| | Learning + optimization | | | | | Optimization | | | | |
|---|---|---|---|---|---|---|---|---|---|---|
| | cora | cite. | prot. | adol | fb | cora | cite. | prot. | adol | fb |
| ClusterNet | **0.54** | **0.55** | **0.29** | **0.49** | **0.30** | **0.72** | **0.73** | **0.52** | **0.58** | 0.76 |
| GCN-e2e | 0.16 | 0.02 | 0.13 | 0.12 | 0.13 | 0.19 | 0.03 | 0.16 | 0.20 | 0.23 |
| Train-CNM | 0.20 | 0.42 | 0.09 | 0.01 | 0.14 | 0.08 | 0.34 | 0.05 | 0.57 | **0.77** |
| Train-Newman | 0.09 | 0.15 | 0.15 | 0.15 | 0.08 | 0.20 | 0.23 | 0.29 | 0.30 | 0.55 |
| Train-SC | 0.03 | 0.02 | 0.03 | 0.23 | 0.19 | 0.09 | 0.05 | 0.06 | 0.49 | 0.61 |
| GCN-2stage-CNM | 0.17 | 0.21 | 0.18 | 0.28 | 0.13 | - | - | - | - | - |
| GCN-2stage-Newman | 0.00 | 0.00 | 0.00 | 0.14 | 0.02 | - | - | - | - | - |
| GCN-2stage-SC | 0.14 | 0.16 | 0.04 | 0.31 | 0.25 | - | - | - | - | - |

Table 2: Performance on the facility location task.

| | Learning + optimization | | | | | Optimization | | | | |
|---|---|---|---|---|---|---|---|---|---|---|
| | cora | cite. | prot. | adol | fb | cora | cite. | prot. | adol | fb |
| ClusterNet | **10** | **14** | **6** | **6** | **4** | 9 | 14 | 6 | 5 | 3 |
| GCN-e2e | 12 | 15 | 8 | **6** | 5 | 11 | 14 | 7 | 6 | 5 |
| Train-greedy | 14 | 16 | 8 | 8 | 6 | 9 | 14 | 7 | 6 | 5 |
| Train-gonzalez | 12 | 17 | 8 | **6** | 6 | 10 | 15 | 7 | 7 | 3 |
| GCN-2Stage-greedy | 14 | 17 | 8 | 7 | 6 | - | - | - | - | - |
| GCN-2Stage-gonzalez | 13 | 17 | 8 | **6** | 6 | - | - | - | - | - |

$\ell = \mathbb{E}_{r^{hard} \sim r}[f(r^{hard}, A^{train})]$, where $r^{hard} \sim r$ denotes this random assignment. $\ell$ is now differentiable in terms of $r$, and can be computed in closed form via standard autodifferentiation tools for many problems of interest (see Section 5). We remark that when the expectation is not available in closed form, our approach could still be applied by repeatedly sampling $r^{hard} \sim r$ and using a policy gradient estimator to compute the gradient of the resulting objective. **(3)** At test time, we simply apply a hard maximum to $r$ to obtain each node's assignment.

**Subset selection:** **(1)** Here, it is less obvious how to obtain a subset of $K$ nodes from the cluster assignments. Our continuous solution will be a vector $x$, $0 \leq x \leq 1$, where $||x||_1 = K$. Intuitively, $x_j$ is the probability of including $x_j$ in the solution. Our approach obtains $x_j$ by placing greater probability mass on nodes that are near the cluster centers. Specifically, each center $\mu_i$ is endowed with one unit of probability mass, which it allocates to the points $x$ as $a_{ij} = \text{softmin}(\eta ||x - \mu_i||)_j$. The total probability allocated to node $j$ is $b_j = \sum_{i=1}^{K} a_{ij}$. Since we may have $b_j > 1$, we pass $b$ through a sigmoid function to cap the entries at 1; specifically, we take $x = 2 * \sigma(\gamma b) - 0.5$ where $\gamma$ is a tunable parameter. If the resulting $x$ exceeds the budget constraint ($||x||_1 > K$), we instead output $\frac{Kx}{||x||_1}$ to ensure a feasible solution.

**(2)** We interpret this solution in terms of the objective similarly as above. Specifically, we consider the result of drawing a discrete solution $x^{hard} \sim x$ where every node $j$ is included (i.e., set to 1) independently with probability $x_j$ from the end of step **(1)**. The training objective is then $\mathbb{E}_{x^{hard} \sim x}[f(x^{hard}, A^{train})]$. For many problems, this can again be computed and differentiated through in closed form (see Section 5).

**(3)** At test time, we need a feasible discrete vector $x$; note that independently rounding the individual entries may produce a vector with more than $K$ ones. Here, we apply a fairly generic approach based on pipage rounding [1], a randomized rounding scheme which has been applied to many problems (particularly those with submodular objectives). Pipage rounding can be implemented to produce a random feasible solution in time $O(n)$ [26]; in practice we round several times and take the solution with the best decision loss on the observed edges. While pipage rounding has theoretical guarantees only for specific classes of functions, we find it to work well even in other domains (e.g., facility location). However, more domain-specific rounding methods can be applied if available.

# 5 Experimental results

We now show experiments on domains that combine link prediction with optimization.

**Learning problem:** In link prediction, we observe a partial graph and aim to infer which unobserved edges are present. In each of the experiments, we hold out $60\%$ of the edges in the graph, with $40\%$ observed during training. We used a graph dataset which is not included in our results to set our method's hyperparameters, which were kept constant across datasets (see appendix for details). The learning task is to use the training edges to predict whether the remaining edges are present, after which we will solve an optimization problem on the predicted graph. The objective is to find a solution with high objective value measured on the *entire* graph, not just the training edges.

**Optimization problems:** We consider two optimization tasks, one from each of the broad classes introduced above. First, *community detection* aims to partition the nodes of the graph into $K$ distinct subgroups which are dense internally, but with few edges across groups. Formally, the objective is to find a partition maximizing the modularity [37], defined as

$$Q(r) = \frac{1}{2m} \sum_{u,v \in V} \sum_{k=1}^{K} \left[ A_{uv} - \frac{d_u d_v}{2m} \right] r_{uk} r_{vk}.$$

Here, $d_v$ is the degree of node $v$, and $r_{vk}$ is 1 if node $v$ is assigned to community $k$ and zero otherwise. This measures the number of edges within communities compared to the expected number if edges were placed randomly. Our clustering module has one cluster for each of the $K$ communities. Defining $B$ to be the modularity matrix with entries $B_{uv} = A_{uv} - \frac{d_u d_v}{2m}$, our training objective (the expected value of a partition sampled according to $r$) is $\frac{1}{2m} \text{Tr} \left[ r^\top B^{train} r \right]$.

Second, minmax *facility location*, where the problem is to select a subset of $K$ nodes from the graph, minimizing the maximum distance from any node to a facility (selected node). Letting $d(v, S)$ be the shortest path length from a vertex $v$ to a set of vertices $S$, the objective is $f(S) = \min_{|S| \leq k} \max_{v \in V} d(v, S)$. To obtain the training loss, we take two steps. First, we replace $d(v, S)$ by $\mathbb{E}_{S \sim x}[d(v, S)]$, where $S \sim x$ denotes drawing a set from the product distribution with marginals $x$. This can easily be calculated in closed form [26]. Second, we replace the $\min$ with a softmin.

**Baseline learning methods:** We instantiate CLUSTERNET using a 2-layer GCN for node embeddings, followed by a clustering layer. We compare to three families of baselines. *First*, GCN-2stage, the two-stage approach which first trains a model for link prediction, and then inputs the predicted graph into an optimization algorithm. For link prediction, we use the GCN-based system of [39] (we also adopt their training procedure, including negative sampling and edge dropout). For the optimization algorithms, we use standard approaches for each domain, outlined below. *Second*, "train", which runs each optimization algorithm only on the observed training subgraph (without attempting any link prediction). *Third*, GCN-e2e, an end-to-end approach which does not include explicit algorithm structure. We train a GCN-based network to directly predict the final decision variable ($r$ or $x$) using the same training objectives as our own model. Empirically, we observed best performance with a 2-layer GCN. This baseline allows us to isolate the benefits of including algorithmic structure.

**Baseline optimization approaches**: In each domain, we compare to expert-designed optimization algorithms found in the literature. In community detection, we compare to "CNM" [11], an agglomerative approach, "Newman", an approach that recursively partitions the graph [36], and "SC", which performs spectral clustering [46] on the modularity matrix. In facility location, we compare to "greedy", the common heuristic of iteratively selecting the point with greatest marginal improvement in objective value, and "gonzalez" [20], an algorithm which iteratively selects the node furthest from the current set. "gonzalez" attains the optimal 2-approximation for this problem (note that the minmax facility location objective is non-submodular, ruling out the usual $(1 - 1/e)$-approximation).

**Datasets:** We use several standard graph datasets: cora [40] (a citation network with 2,708 nodes), citeseer [40] (a citation network with 3,327 nodes), protein [14] (a protein interaction network with 3,133 nodes), adol [12] (an adolescent social network with 2,539 vertices), and fb [13, 32] (an online social network with 2,888 nodes). For facility location, we use the largest connected component of the graph (since otherwise distances may be infinite). Cora and citeseer have node features (based on

Table 3: Inductive results. "%" is the fraction of test instances for which a method attains top performance (including ties). "Finetune" methods are excluded from this in the "No finetune" section.

| | Community detection | | | | | Facility location | | | |
| | synthetic | | pubmed | | | synthetic | | pubmed | |
| No finetune | Avg. | % | Avg. | % | No finetune | Avg. | % | Avg. | % |
|---|---|---|---|---|---|---|---|---|---|
| ClusterNet | **0.57** | **26/30** | **0.30** | **7/8** | ClusterNet | **7.90** | **25/30** | 7.88 | 3/8 |
| GCN-e2e | 0.26 | 0/30 | 0.01 | 0/8 | GCN-e2e | 8.63 | 11/30 | 8.62 | 1/8 |
| Train-CNM | 0.14 | 0/30 | 0.16 | 1/8 | Train-greedy | 14.00 | 0/30 | 9.50 | 1/8 |
| Train-Newman | 0.24 | 0/30 | 0.17 | 0/8 | Train-gonzalez | 10.30 | 2/30 | 9.38 | 1/8 |
| Train-SC | 0.16 | 0/30 | 0.04 | 0/8 | 2Stage-greedy | 9.60 | 3/30 | 10.00 | 0/8 |
| 2Stage-CNM | 0.51 | 0/30 | 0.24 | 0/8 | 2Stage-gonz. | 10.00 | 2/30 | **6.88** | **5/8** |
| 2Stage-Newman | 0.01 | 0/30 | 0.01 | 0/8 | ClstrNet-1train | 7.93 | 12/30 | 7.88 | 2/8 |
| 2Stage-SC | 0.52 | 4/30 | 0.15 | 0/8 | | | | | |
| ClstrNet-1train | 0.55 | 0/30 | 0.25 | 0/8 | | | | | |
| Finetune | | | | | Finetune | | | | |
| ClstrNet-ft | 0.60 | 20/30 | 0.40 | 2/8 | ClstrNet-ft | 8.08 | 12/30 | 8.01 | 3/8 |
| ClstrNet-ft-only | 0.60 | 10/30 | 0.42 | 6/8 | ClstrNet-ft-only | 7.84 | 16/30 | 7.76 | 4/8 |

a bag-of-words representation of the document), which were given to all GCN-based methods. For the other datasets, we generated unsupervised node2vec features [23] using the training edges.

## 5.1 Results on single graphs

We start out with results for the combined link prediction and optimization problem. Table 1 shows the objective value obtained by each approach on the full graph for community detection, with Table 2 showing facility location. We focus first on the "Learning + Optimization" column which shows the combined link prediction/optimization task. We use $K = 5$ clusters; $K = 10$ is very similar and may be found in the appendix. CLUSTERNET outperforms the baselines in nearly all cases, often substantially. GCN-e2e learns to produce nontrivial solutions, often rivaling the other baseline methods. However, the explicit structure used by our approach CLUSTERNET results in much higher performance.

Interestingly, the two stage approach sometimes performs worse than the train-only baseline which optimizes just based on the training edges (without attempting to learn). This indicates that approaches which attempt to accurately reconstruct the graph can *sometimes* miss qualities which are important for optimization, and in the worst case may simply add noise that overwhelms the signal in the training edges. In order to confirm that the two-stage method learned to make meaningful predictions, in the appendix we give AUC values for each dataset. The average AUC value is 0.7584, indicating that the two-stage model does learn to make nontrivial predictions. However, the small amount of training data (only 40% of edges are observed) prevents it from perfectly reconstructing the true graph. This drives home the point that decision-focused learning methods such as CLUSTERNET can offer substantial benefits when highly accurate predictions are out of reach even for sophisticated learning methods.

We next examine an optimization-only task where the entire graph is available as input (the "Optimization" column of Tables 1 and Table 2). This tests CLUSTERNET's ability to learn to solve combinatorial optimization problems compared to expert-designed algorithms, even when there is no partial information or learning problem in play. We find that CLUSTERNET is highly competitive, meeting and frequently exceeding the baselines. It is particularly effective for community detection, where we observe large (> 3x) improvements compared to the best baseline on some datasets. At facility location, our method always at least ties the baselines, and frequently improves on them. These experiments provide evidence that our approach, which is automatically specialized during training to optimize on a given graph, can rival and exceed hand-designed algorithms from the literature. The alternate learning approach, GCN-e2e, which is an end-to-end approach that tries to learn to predicts optimization solutions directly from the node features, at best ties the baselines and typically underperforms. This underscores the benefit of including algorithmic structure as a part of the end-to-end architecture.

## 5.2 Generalizing across graphs

Next, we investigate whether our method can learn generalizable strategies for optimization: can we train the model on one set of graphs drawn from some distribution and then apply it to unseen graphs? We consider two graph distributions. First, a synthetic generator introduced by [48], which is based on the spatial preferential attachment model [7] (details in the appendix). We use 20 training graphs, 10 validation, and 30 test. Second, a dataset obtained by splitting the pubmed graph into 20 components using metis [27]. We fix 10 training graphs, 2 validation, and 8 test. At test time, only 40% of the edges in each graph are revealed, matching the "Learning + optimization" setup above.

Table 3 shows the results. To start out, we do not conduct any fine-tuning to the test graphs, evaluating entirely the generalizability of the learned representations. CLUSTERNET outperforms all baseline methods on all tasks, except for facility location on pubmed where it places second. We conclude that the learned model successfully generalizes to completely unseen graphs. We next investigate (in the "finetune" section of Table 3) whether CLUSTERNET's performance can be further improved by fine-tuning to the 40% of observed edges for each test graph (treating each test graph as an instance of the link prediction problem from Section 5.1, but initializing with the parameters of the model learned over the training graphs). We see that CLUSTERNET's performance typically improves, indicating that fine-tuning can allow us to extract additional gains if extra training time is available.

Interestingly, *only* fine-tuning (not using the training graphs at all) yields similar performance (the row "ClstrNet-ft-only"). While our earlier results show that CLUSTERNET can learn generalizable strategies, doing so may not be necessary when there is the opportunity to fine-tune. This allows a trade-off between quality and runtime: without fine-tuning, applying our method at test time requires just a single forward pass, which is extremely efficient. If additional computational cost at test time is acceptable, fine-tuning can be used to improve performance. Complete runtimes for all methods are shown in the appendix. CLUSTERNET's forward pass (i.e., no fine-tuning) is extremely efficient, requiring at most 0.23 seconds on the largest network, and is *always faster than the baselines* (on identical hardware). Fine-tuning requires longer, on par with the slowest baseline.

We lastly investigate the reason why pretraining provides little to no improvement over only fine-tuning. Essentially, we find that CLUSTERNET is extremely sample-efficient: using only a single training graph results in nearly as good performance as the full training set (and still better than all of the baselines), as seen in the "ClstrNet-1train" row of Table 3. *That is,* CLUSTERNET *is capable of learning optimization strategies that generalize with strong performance to completely unseen graphs after observing only a single training example*. This underscores the benefits of including algorithmic structure as a part of the architecture, which guides the model towards learning meaningful strategies.

## 6 Conclusion

When machine learning is used to inform decision-making, it is often necessary to incorporate the downstream optimization problem into training. Here, we proposed a new approach to this decision-focused learning problem: include a differentiable solver for a simple proxy to the true, difficult optimization problem and learn a representation that maps the difficult problem to the simpler one. This representation is trained in an entirely automatic way, using the solution quality for the true downstream problem as the loss function. We find that this "middle path" for including algorithmic structure in learning improves over both two-stage approaches, which separate learning and optimization entirely, and purely end-to-end approaches, which use learning to directly predict the optimal solution. Here, we instantiated this framework for a class of graph optimization problems. We hope that future work will explore such ideas for other families of problems, paving the way for flexible and efficient optimization-based structure in deep learning.

## Acknowledgements

This work was supported by the Army Research Office (MURI W911NF1810208). Wilder is supported by a NSF Graduate Research Fellowship. Dilkina is supported partially by NSF award # 1914522 and by U.S. Department of Homeland Security under Grant Award No. 2015-ST-061-CIRC01. The views and conclusions contained in this document are those of the authors and should not be interpreted as necessarily representing the official policies, either expressed or implied, of the U.S Department of Homeland Security.

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
