[Supplementary Material · appendix.pdf]

# Appendix: End to end learning and optimization on graphs

## 1 Proofs

### 1.1 Exact expression for gradients

Define $R_i = \sum_{j=1}^{n} r_{ji}$ and $C_i = \sum_{j=1}^{n} r_{ji} x_j$. We will work with $C_i \in R^{p \times 1}$ as a column vector. For a fixed $i, j$, we have

$$\frac{\partial f_{i,\cdot}}{\partial x_j} = -\frac{R_i x_j \left[\frac{\partial r_{ji}}{\partial x_j}\right]^{\top} - C_i \left[\frac{\partial r_{ji}}{\partial x_j}\right]^{\top}}{R_i^2} - \frac{r_{ji}}{R_i} I$$

where $I$ denotes the $p$-dimensional identity matrix. Similarly, fixing $i, k$ gives

$$\frac{\partial f_{i,\cdot}}{\partial \mu_k} = \delta_{ik} I - \frac{R_i \sum_{j=1}^{n} x_j \left[\frac{\partial r_{ji}}{\partial \mu_k}\right]^{\top} - C_i \left[\sum_{j=1}^{n} \frac{\partial r_{ji}}{\partial \mu_k}\right]^{\top}}{R_i^2}$$

### 1.2 Guarantee for approximate gradients

**Proposition 1.** *Suppose that for all points $j$, $||x_j - \mu_i|| - ||x_j - \mu_{c(j)}|| \geq \delta$ for all $i \neq c(j)$ and that for all clusters $i$, $\sum_{j=1}^{n} r_{ji} \geq \alpha n$. Moreover, suppose that $\beta\delta > \log \frac{2\beta K^2}{\alpha}$. Then, $\left\|\left|\frac{\partial f}{\partial \mu} - I\right\|\right|_1 \leq$ $\exp(-\delta\beta)\left(\frac{K^2\beta}{\frac{1}{2}\alpha - K^2\beta \exp(-\delta\beta)}\right)$ where $|| \cdot ||_1$ is the operator 1-norm.*

We focus on the off-diagonal component of $\frac{\partial f_{im}}{\partial \mu_{k\ell}}$, given by

$$A_{(i,m),(k,\ell)} = -\frac{R_i \sum_{j=1}^{n} x_j^m \left[\frac{\partial r_{ji}}{\partial \mu_k^\ell}\right] - C_i^m \left[\sum_{j=1}^{n} \frac{\partial r_{ji}}{\partial \mu_k^\ell}\right]}{R_i^2}$$

The key term here is $\frac{\partial r_{ji}}{\partial \mu_k^\ell}$. Let $s_{ji} = -\beta ||x_j - \mu_i||$ Since $r$ is defined via the softmax function, we have

$$\frac{\partial r_{ji}}{\partial \mu_k^\ell} = \frac{\partial r_{ji}}{\partial s_{jk}} \frac{\partial s_{jk}}{\partial \mu_k^\ell}$$

where

$$\frac{\partial r_{ji}}{\partial s_{jk}} = \begin{cases} r_{ji}(1 - r_{ji}) & \text{if i = k} \\ -r_{ji}r_{jk} & \text{otherwise.} \end{cases}$$

Note now via Lemma 1, in both cases we have that

$$\left|\frac{\partial r_{ji}}{\partial s_{jk}}\right| \le K \exp(-\beta\delta)$$

Define $\epsilon = K \exp(-\beta\delta)$ and note that we have that $\left|\frac{\partial s_{jk}}{\partial \mu_k^\ell}\right| \le \beta$, since we defined $s$ in terms of cosine similarity and have assumed that the input is normalized. Putting this together, we have

$$\left|A_{(i,m),(k,\ell)}\right| \le \frac{\sum_{j=1}^n x_j^m \epsilon\beta}{R_i} + \frac{C_i^m n\epsilon\beta}{R_i^2}$$
$$\le \frac{\epsilon\beta \sum_{j=1}^n x_j^m}{\alpha n} + \frac{\mu_i^m n\epsilon\beta}{R_i}$$
$$\le \frac{\epsilon\beta \sum_{j=1}^n x_j^m}{\alpha n} + \frac{\mu_i^m \epsilon\beta}{\alpha}$$

and so

$$||A||_1 = \max_{(k,\ell)} \sum_{(i,m)} A_{(i,m),(k,\ell)}$$
$$\le \max_{(k,\ell)} \sum_{(i,m)} \frac{\epsilon\beta \sum_{j=1}^n x_j^m}{\alpha n} + \frac{\mu_i^m \epsilon\beta}{\alpha}$$
$$\le \max_{(k,\ell)} \sum_i \frac{\epsilon\beta n}{\alpha n} + \frac{\epsilon\beta}{\alpha} \quad (\text{since } ||x_j||_1, ||\mu_i||_1 \le 1)$$
$$\le \frac{2K\epsilon\beta}{\alpha}$$
$$= \frac{2K^2\beta \exp(-\beta\delta)}{\alpha}.$$

Since by assumption $\beta\delta > \log \frac{2\beta K^2}{\alpha}$, we know that $||A||_1 < 1$ and applying Lemma 2 competes the proof.

**Lemma 1.** *Consider a point $j$ and let $i = \arg\max_k r_{jk}$. Then, $r_{ji} \ge \frac{1}{1+K\exp(-\beta\delta)}$, and correspondingly, $\sum_{k\neq i} r_{jk} \le \frac{K\exp(-\beta\delta)}{K\exp(-\beta\delta)+1} \le K\exp(-\beta\delta)$.*

*Proof.* Equation 4 of [2] gives that

$$r_{ij} \ge \prod_{k\neq i} \frac{1}{1 + \exp(-(s_i - s_k))}.$$

Since by assumption we have $-||x_j - \mu_i|| \geq \delta||x_j - \mu_k||$, we obtain

$$r_{ij} \geq \prod_{k \neq i} \frac{1}{1 + \exp(-\delta\beta)}$$

$$\geq \frac{1}{1 + K\exp(-\delta\beta)} \quad \text{(using that } \exp(-\delta\beta) \leq 1).$$

which proves the lemma. $\qquad\square$

**Lemma 2.** *Suppose that for a matrix $A$, $||A - I|| \leq \delta$ for some $\delta < 1$ and an operator norm $|| \cdot ||$. Then, $||A^{-1} - I|| \leq \frac{\delta}{1-\delta}$.*

*Proof.* Let $B = I - A$. We have

$$A^{-1} = (I - B)^{-1}$$

$$= \sum_{i=0}^{\infty} B^i \quad \text{(using the Neumann series representation)}$$

$$= I + \sum_{i=1}^{\infty} B^i$$

and so $||A^{-1} - I||_{\infty} = \left|\left|\sum_{i=1}^{\infty} B^i\right|\right|_{\infty}$. We have

$$\left|\left|\sum_{i=1}^{\infty} B^i\right|\right|_{\infty} \leq \sum_{i=1}^{\infty} ||B^i||_{\infty}$$

$$\leq \sum_{i=1}^{\infty} ||B||_{\infty}^i \quad \text{(since operator norms are submultiplicative)}$$

$$= \frac{\delta}{1 - \delta} \quad \text{(geometric series).}$$

$\qquad\square$

# 2 Experimental setup details

## 2.1 Hyperparameters

All methods were trained with the Adam optimizer. For the single-graph experiments, we tested the following settings on the pubmed graph (which was not used in our single-graph experiments):

- $\beta = 1, 10, 30, 50$

- learning rate = 0.01, 0.001

- training iterations = 100, 200, ..., 1000

- Number of forward pass $k$-means updates: 1, 3

- Whether to increase the number of $k$-means updates to 5 after 500 training iterations.

- GCN hidden layer size: 20, 50, 100

- Embedding dimension: 20, 50, 100

For all single-graph experiments, we used $\beta = 30$, $\gamma = 100$, GCN hidden layer = embedding dimension = 50, 1 $k$-means update in the forward pass, learning rate = 0.01, and 1000 training iterations, with the number of $k$-means updates increasing to 5 after 500 iterations.

We tested the following set of hyperparameters on the validation set for each graph distribution

- $\beta = 30, 50, 70, 100$

- learning rate = 0.01, 0.001

- dropout = 0.5, 0.2

- training iterations = 10, 20...300

- Number of forward pass $k$-means updates: 1, 5, 10, 15

- Hidden layer size: 20, 50, 100

- Embedding dimension: 20, 50, 100

We selected $\beta = 70$, learning rate = 0.001, dropout = 0.2, and hidden layer = embedding dimension = 50 for all experiments. On the synthetic graphs we used 70 training iterations and 10 forward-pass $k$-means updates. For pubmed, we used 220 and 1, respectively.

## 2.2   Synthetic graph generation

Each node has a set of attributes $y_i$ (in this case, demographic features simulated from real population data); node $i$ forms a connection to node $j$ with probability proportional to $e^{-\frac{1}{\rho}||y_i - y_j||}d(j)$ where $d(j)$ is the degree of node $j$. This models both the homophily and heavy-tailed degree distribution seen in real world networks. We took $\rho = 0.025$ to obtain a high degree of homophily, so that there is meaningful community structure. In order to make the problem more difficult, our method does not observe the features $y$; instead, we generate unsupervised features from the graph structure alone using role2vec [1] (which generates inductive representations based on motif counts that are meaningful across graphs). Each graph has 500 nodes.

## 2.3   Code

The code used for the experiments is included in the supplemental material. Due to file size limitations, the data can be found at `https://www.dropbox.com/s/5ru0xyzojdk7wn8/data_graphopt.zip?dl=0`. The files "singlegraph_linkpred.py" and "distributional_linkpred.py" run the experiments or the single graph and inductive settings, respectively. We use PyTorch version 1.1.0. All GCNs are implemented using the pygcn package (`https://github.com/tkipf/pygcn`). Networkx version 2.3 is used for many graph operations, along with igraph 0.7.1 to accelerate shortest path computations. The included Dockerfile builds the environment we used for the experiments, with the exception of pygcn, which must be downloaded from github and installed separately.

## 2.4 Hardware

All methods were run on a machine with 14 i9 3.1 GHz cores and 128 GB of RAM. For fair runtime comparisons with the baselines, all methods were run on CPU.

# 3 Results for $K = 10$

Table 1: Results for community detection. "-" for GCN-2Stage-Newman in the Learning + optimization section denotes that the method could not be run due to numerical issues.

|  | Learning + optimization | | | | | Optimization | | | | |
|---|---|---|---|---|---|---|---|---|---|---|
|  | cora | cite. | prot. | adol | fb | cora | cite. | prot. | adol | fb |
| ClusterNet | **0.56** | **0.53** | **0.28** | **0.47** | **0.28** | **0.71** | **0.76** | **0.52** | 0.55 | **0.80** |
| GCN-e2e | 0.01 | 0.01 | 0.06 | 0.08 | 0.00 | 0.07 | 0.08 | 0.14 | 0.15 | 0.15 |
| Train-CNM | 0.20 | 0.44 | 0.09 | 0.01 | 0.17 | 0.08 | 0.34 | 0.05 | **0.60** | **0.80** |
| Train-Newman | 0.08 | 0.15 | 0.15 | 0.14 | 0.07 | 0.20 | 0.22 | 0.29 | 0.30 | 0.47 |
| Train-SC | 0.06 | 0.04 | 0.05 | 0.22 | 0.21 | 0.15 | 0.08 | 0.07 | 0.46 | 0.79 |
| GCN-2stage-CNM | 0.20 | 0.23 | 0.18 | 0.32 | 0.08 | - | - | - | - | - |
| GCN-2stage-Newman | 0.01 | 0.00 | 0.00 | - | 0.00 | - | - | - | - | - |
| GCN-2stage-SC | 0.13 | 0.18 | 0.10 | 0.29 | 0.18 | - | - | - | - | - |

Table 2: Results for facility location

|  | Learning + optimization | | | | | Optimization | | | | |
|---|---|---|---|---|---|---|---|---|---|---|
|  | cora | cite. | prot. | adol | fb | cora | cite. | prot. | adol | fb |
| ClusterNet | **9** | **14** | **7** | **5** | **2** | 8 | 13 | 6 | 5 | **2** |
| GCN-e2e | 12 | 15 | 8 | 6 | 4 | 10 | 14 | 7 | **5** | 4 |
| Train-greedy | 14 | 16 | 8 | 8 | 6 | 9 | 14 | 7 | 6 | 5 |
| Train-gonzalez | 11 | 15 | 8 | 7 | 6 | 9 | **13** | 7 | 6 | **2** |
| GCN-2Stage-greedy | 14 | 16 | 8 | 7 | 6 | - | - | - | - | - |
| GCN-2Stage-gonzalez | 12 | 16 | 8 | 6 | 5 | - | - | - | - | - |

# 4 Timing Results

We run experiments on Intel i9 7940X @ 3.1 GHz with 128 GB of RAM. We report runtime in seconds. For algorithms with learned models, we report both the training time and the time to complete a single forward pass.

Table 3: Timing results for the community detection task (s)

|  | cora | cite. | prot. | adol | fb |
|---|---|---|---|---|---|
| ClusterNet - Training Time | 59.48 | 149.73 | 129.63 | 56.68 | 54.33 |
| ClusterNet - Forward Pass | 0.04 | 0.12 | 0.11 | 0.04 | 0.05 |
| GCN-e2e - Training Time | 36.83 | 54.99 | 34.60 | 29.04 | 28.17 |
| GCN-e2e - Forward Pass | 0.002 | 0.005 | 0.002 | 0.003 | 0.001 |
| Train-CNM | 1.31 | 1.28 | 1.02 | 1.03 | 2.94 |
| Train-Newman | 9.99 | 15.89 | 15.19 | 11.45 | 7.25 |
| Train-SC | 0.41 | 0.62 | 0.55 | 0.38 | 0.48 |
| GCN-2Stage - Training Time | 68.79 | 72.20 | 75.69 | 103.56 | 57.62 |
| GCN-2Stage-CNM | 119.34 | 178.39 | 159.64 | 101.64 | 142.02 |
| GCN-2Stage-New. | 37.96 | 58.26 | 51.70 | 33.14 | 43.88 |
| GCN-2Stage-SC | 0.40 | 0.61 | 0.50 | 0.33 | 0.36 |

Table 4: Timing results for the kcenter task (s)

|  | cora | cite. | prot. | adol | fb |
|---|---|---|---|---|---|
| ClusterNet - Training Time | 264.14 | 555.84 | 488.37 | 244.74 | 246.57 |
| ClusterNet - Forward Pass | 0.10 | 0.23 | 0.20 | 0.09 | 0.11 |
| GCN-e2e - Training Time | 237.68 | 511.23 | 446.76 | 229.49 | 221.28 |
| GCN-e2e - Forward Pass | 0.003 | 0.006 | .005 | 0.004 | .003 |
| Train-Greedy | 1029.18 | 2387 | 1966 | 619.06 | 1244.09 |
| Train-Gonzalez | 0.082 | 0.14 | 0.12 | 0.07 | .066 |
| GCN-2Stage - Training Time | 73.82 | 70.21 | 103.98 | 75.48 | 104.66 |
| GCN-2Stage-Greedy | 1189.15 | 2367 | 2017 | 621.59 | 1237.871 |
| GCN-2Stage-Gonzalez | 0.18 | 0.28 | 0.25 | 0.13 | 0.13 |

Table 5: Timing results in the inductive setting for community detection task (s)

|  | synthetic | pubmed |
|---|---|---|
| ClusterNet - Training time | 6.57 | 13.74 |
| ClusterNet - Forward Pass | 0.003 | 0.008 |
| GCN-e2e - Training time | 11.40 | 15.86 |
| GCN-e2e - Forward Pass | 0.04 | 0.03 |
| Train-CNM | 0.08 | 0.17 |
| Train-Newman | 0.65 | 1.83 |
| Train-SC | 0.03 | 0.04 |
| 2Stage - Train | 10.98 | 15.86 |
| 2Stage-CNM | 3.23 | 13.73 |
| 2Stage-New. | 1.12 | 4.29 |
| 2Stage-SC | 0.04 | 0.10 |

Table 6: Timing results in the inductive setting for the kcenter task (s)

|  | synthetic | pubmed |
|---|---|---|
| ClusterNet - Training Time | 14.36 | 43.06 |
| ClusterNet - Forward Pass | 0.005 | 0.02 |
| GCN-e2e - Training Time | 9.49 | 33.73 |
| GCN-e2e - Forward Pass | 0.01 | 0.02 |
| Train-Gonzalez | 0.07 | 0.49 |
| Train-Greedy | 4.99 | 32.7 |
| 2Stage - Train | 11.00 | 15.78 |
| 2Stage-Gonzalez | 0.07 | 0.07 |
| 2Stage-Greedy | 5.31 | 16.16 |