[Reviews · NeurIPS 2019]

Reviewer 1



Usually research on graph embeddings will first learn general-purpose graph embeddings. Then the learned embeddings will be used to solve various tasks regarding the graph. This paper notices that if we combine embedding learning and task optimization in an end-to-end pipeline, better results can be achieved for the tasks. The proposed method makes intuitive sense. Instead of learning general embeddings, the method basically learns task-specific embeddings. It is not surprising that better empirical results can be achieved. But I would argue that the proposed method sacrifices the ability to learn general-purpose embeddings. The new embeddings not longer capture general properties of the graph, but just properties related to whatever optimization task at hand, such as clustering. Also, because the embeddings are no longer general-purpose, I wonder whether it is still necessary to have the explicit graph embedding layer. Would it be possible to just have a single system specifically designed for the task optimization?

Reviewer 2



In general, the idea of this paper is pretty simple and neat. Although there are closely related ideas in the literature (and the authors cited those), this paper delivers a completely differentiable approach to the problems it is to solve. - As this paper has promised, it only works for those combinatorial problems that involve a hidden grouping or (its dual) partitioning. Many other combinatorial problems on graphs remain untouched. - They rely on efficient computation of the expected training loss, $\ell$. Although one can compute such loss for a group of problems, it is not likely to compute such an expectation in every case. It is not clear how one can approximate such differentials. - Line 243: no need for the Trace. - Line 267: the 2-approximation is not optimal, as it is (1 - 1/e) for Facility Location. - The intuitions on Section 5 are nice but are not backed up with mathematical reasoning. It is hard to reason about DNNs in general, though.

Reviewer 3



This work builds on a strong foundation of related work; but does so in a novel way. The paper is very clearly written, which is much appreciated! As for quality, the authors perform a thorough analysis comparing against a variety of expert baselines on a variety of data sets. Their supplemental material is very comprehensive. As for significance, NeurIPS attendees will be interested in the pipeline presented and the novel incorporation of a differentiable proxy graph optimization; but they might not be as interested in the problem of graph optimization more broadly.

[Author Response · NeurIPS 2019]

We thank the reviewers for their comments, and will incorporate their suggestions to improve the paper.

**Response to Reviewer 1:** Regarding task-specific vs general-purpose embeddings: there is an important role for both kinds of techniques. General-purpose embeddings may be valuable for discovering properties of the graph. However, state of the art performance for any given problem almost always requires fine-tuning the embeddings for a particular task (e.g., recent results for link prediction [6, 7] or semi-supervised classification [4, 2]). Our contribution is a way of achieving effective end-to-end embeddings for substantially more complex problems involving discrete optimization.

Regarding whether an embedding layer is necessary: to our knowledge, all modern learning-based systems for discrete optimization tasks (e.g., [3, 1, 5]) first embed any discrete inputs into a continuous domain in order to harness the power of deep networks and gradient-based training. In some cases, this embedding has been trivial because only Euclidean graphs were considered [1, 5], while others used explicit embedding layers to handle arbitrary graph structures [3]. The motivation for our architecture is to allow embeddings for arbitrary graphs to be customized to combined learning + optimization tasks.

**Response to Reviewer 2:** Regarding the range of problems we consider: we remark that the set of problems in the paper already span a wide range of algorithm design paradigms: we compare to spectral methods, greedy maximization, recursive partitioning schemes, etc. The overarching problem classes that our method applies to also span a wide range of applications: community detection, maxcut, facility location, influence maximization, and immunization problems, just to name a few examples.

However, our method is not limited just to the problem classes considered in the paper; any problems where the clustering layer's output can be interpreted as a soft solution is eligible. Exploring additional applications of our method would be an interesting topic.

Regarding closed-form loss functions: we selected this loss function, based on independently rounding each coordinate, specifically because it often results in a closed-form loss (i.e., this is a deliberate advantage to our approach). However, in cases where the loss is not available in closed form, it is always possible to draw samples from the independent rounding scheme and apply the REINFORCE estimator. Each sample only requires evaluating the objective function, which is typically cheap relative to backpropagation.

**Response to Reviewer 3:** Thanks for your comments! If accepted, we would use part of the additional page to add a conclusion.

# References

[1] Irwan Bello, Hieu Pham, Quoc V Le, Mohammad Norouzi, and Samy Bengio. Neural combinatorial optimization with reinforcement learning. *arXiv preprint arXiv:1611.09940*, 2016.

[2] W. Hamilton, Z. Ying, and J. Leskovec. Inductive representation learning on large graphs. In *NIPS*, 2017.

[3] E. Khalil, H. Dai, Y. Zhang, B. Dilkina, and L. Song. Learning combinatorial optimization algorithms over graphs. In *NIPS*, 2017.

[4] T. Kipf and M. Welling. Semi-supervised classification with graph convolutional networks. In *ICLR*, 2017.

[5] Wouter Kool, Herke van Hoof, and Max Welling. Attention, learn to solve routing problems! In *ICLR*, 2019.

[6] M. Schlichtkrull, T. Kipf, P. Bloem, R. Van Den Berg, I. Titov, and M. Welling. Modeling relational data with graph convolutional networks. In *European Semantic Web Conference*, 2018.

[7] M. Zhang and Y. Chen. Link prediction based on graph neural networks. In *NIPS*, 2018.


[Meta-Review · NeurIPS 2019]

* Introduces a GNN layer that fits in between OptNet and generic GNNs where k-means is run in the forward pass and the backward pass uses implicit differentiation. The architecture is well-suited to problems with latent cluster-based structure. Also introduces differentiable relaxations of problem-specific decision-theoretic losses & a nice approximate implicit derivative that speeds up computation through the k-means fixed point. * Reviewers are generally positive, though AC wasn’t particularly happy with the depth of the reviews and went through the paper in detail. The paper is good quality and the new approximation for differentiating through k-means may have more general applicability. However AC has the following concerns about related work: 1. Hierarchical Graph Representation Learning with Differentiable Pooling (NeurIPS 2018, https://arxiv.org/abs/1806.08804). Specifically look at Eq 3-4 in this paper, where r from the submission corresponds to S mu from the submission corresponds to X. This is a simpler alternative way of producing the assignment probabilities and centroid vectors within a GNN architecture. AC would have appreciated a baseline that replaced the k-means step in the submission with this step, and then followed with the same loss function. 2. Generalized approximate graph partitioning gives a differentiable proxy of the normalized cut objective: https://arxiv.org/abs/1903.00614 3. Could also cite https://arxiv.org/abs/1803.06396 and the original Almeida-Pineda works (see abstract therein) in the context of the implicit function theorem. Should discuss around L176-189 that it’s possible to run an iterative algorithm in the backward pass to compute the implicit gradient without storing the forward pass (see Almeida-Pineda ’87). It doesn’t have the drawbacks of differentiating through the unrolled algorithm discussed there. * We discussed this in the discussion phase, and the reviewers felt it would be nice but didn't change their recommendation of acceptance. AC is ok with that, but the authors should add discussion of these works, and would encourage them to experimentally compare to 1 for the final version.